# Developing a multivariable prediction model to support personalized selection among five major empirically-supported treatments for adult depression. Study protocol of a systematic review and individual participant data network meta-analysis

Ellen Driessen[1,2]*, Orestis Efthimiou[3,4,5], Frederik J. Wienicke[2], Jasmijn Breunese[2], Pim Cuijpers[6,7], Thomas P. A. Debray[8,9], David J. Fisher[10], Marjolein Fokkema[11], Toshiaki A. Furukawa[12], Steven D. Hollon[13], Anuj H. P. Mehta[14], Richard D. Riley[15], Madison R. Schmidt[16], Jos W. R. Twisk[17], Zachary D. Cohen[18]*

1 Department of Clinical Psychology, Behavioural Science Institute, Radboud University, Nijmegen, Netherlands, 2 Depression Expertise Center, Pro Persona Mental Health Care, Nijmegen, Netherlands, 3 Institute of Social and Preventive Medicine, University of Bern, Bern, Switzerland, 4 Institute of Primary Health Care, University of Bern, Bern, Switzerland, 5 Department of Psychiatry, University of Oxford, Oxford, United Kingdom, 6 Department of Clinical, Neuro and Developmental Psychology, Amsterdam Public Health Research Institute, Vrije Universiteit Amsterdam, Amsterdam, Netherlands, 7 International Institute for Psychotherapy, Babeș-Bolyai University, Cluj-Napoca, Romania, 8 Smart Data Analysis and Statistics Besloten Vennootschap, Utrecht, Netherlands, 9 Julius Center for Health Sciences and Primary Care, University Medical Center Utrecht, Utrecht, Netherlands, 10 Medical Research Council Clinical Trials Unit, Institute of Clinical Trials and Methodology, University College London, London, United Kingdom, 11 Department of Methods and Statistics, Institute of Psychology, Leiden University, Leiden, Netherlands, 12 Department of Health Promotion and Human Behavior, Department of Clinical Epidemiology, Kyoto University Graduate School of Medicine/School of Public Health, Kyoto, Japan, 13 Department of Psychology, Vanderbilt University, Nashville, United States of America, 14 Department of Psychological and Brain Sciences, University of Massachusetts Amherst, Amherst, United States of America, 15 Institute of Applied Health Research, University of Birmingham, Birmingham, United Kingdom, 16 Department of Clinical Psychology, Northwestern University Chicago, Chicago, United States of America, 17 Department of Epidemiology and Data Science, Amsterdam University Medical Centers, Amsterdam, Netherlands, 18 Department of Psychology, University of Arizona, Tucson, United States of America

* ellen.driessen@ru.nl (ED); zachary.d.cohen@gmail.com (ZDC)

## Abstract

### Background

Various treatments are recommended as first-line options in practice guidelines for depression, but it is unclear which is most efficacious for a given person. Accurate individualized predictions of relative treatment effects are needed to optimize treatment recommendations for depression and reduce this disorder's vast personal and societal costs.

### Aims

We describe the protocol for a systematic review and individual participant data (IPD) network meta-analysis (NMA) to inform personalized treatment selection among five major empirically-supported depression treatments.

**Data availability statement:** Data availability is not applicable to this article as no data were analyzed for this study protocol paper.

**Funding:** ED received funding from the Dutch Research Council (https://www.nwo.nl/en) to supported this work (Grant number: NWO 016. Veni.195.215 6806). DF is supported by UK Research and Innovation Medical Research Council (https://www.ukri.org/councils/mrc/), grant number: MC_UU_00004/06. The funders had no role in the development of this study protocol, nor was there editorial direction or censorship from the sponsor in this manuscript.

**Competing interests:** I have read the journal's policy and the authors of this manuscript have the following competing interests: Dr. Driessen reports a grant from the Dutch Research Council during the conduct of the study. Dr. Debray owns a company that provides consulting services to pharmaceutical companies and contract research organizations. Dr. Debray is also editing a book on comparative effectiveness and personalized medicine research, for which he will receive royalties upon publication. Dr. Furukawa reports personal fees from Boehringer-Ingelheim, DT Axis, Kyoto University Original, Shionogi and SONY, and a grant from Shionogi, outside the submitted work. In addition, Dr. Furukawa has patents 2020-548587 and 2022-082495 pending, and intellectual properties for Kokoro-app licensed to Mitsubishi-Tanabe. Dr. Cohen previously held stock options in AbleTo, which he received as compensation for advice on the clinical content of Joyable's digital cognitive therapy for depression. The remaining authors have nothing to declare.

## Method

We will use the METASPY database to identify randomized clinical trials that compare two or more of five treatments for adult depression: antidepressant medication, cognitive therapy, behavioral activation, interpersonal psychotherapy, and psychodynamic therapy. We will request IPD from identified studies. We will conduct an IPD-NMA and develop a multivariable prediction model that estimates individualized relative treatment effects from demographic, clinical, and psychological participant characteristics. Depressive symptom level at treatment completion will constitute the primary outcome. We will evaluate this model using a range of measures for discrimination and calibration, and examine its potential generalizability using internal-external cross-validation.

## Conclusions

We describe a state-of-the-art method to predict personalized treatment effects based on IPD from multiple trials. The resulting prediction model will need prospective evaluation in mental health care for its potential to inform shared decision-making. This study will result in a unique database of IPD from randomized clinical trials around the world covering five widely used depression treatments, available for future research.

## 1. Introduction

People suffering from depression have a range of therapeutic options, including various antidepressant medications (ADMs) and psychological treatments such as cognitive therapy (CT), behavioral activation (BA), interpersonal psychotherapy (IPT), and short-term psychodynamic psychotherapy (STPP). Although no significant differences in average treatment effects have been found between these interventions [1,2], response is highly heterogeneous [3–5] warranting the need for more personalized treatment recommendations [6].

Current scientific methodologies are limited in their ability to discriminate individual response. What little information is available regarding patient characteristics associated with differential efficacy of depression treatments has been mostly obtained from univariable analyses [7]. These studies typically examine how an isolated patient characteristic (e.g., age) predicts differential outcomes across two treatment groups (e.g., ADM versus CT) [8–11]. Unfortunately, single predictors tend to have small effects [12], and their investigation is prone to false positive results and selective reporting. Moreover, clinical reality is much more complex, involving numerous patient characteristics and contextual factors, whose associations can vary across different treatment comparisons. Identifying and combining multiple prescriptive predictors is crucial for clinical decision-making [12–14]. Recent efforts in this regard for specific head-to-head treatment comparisons have shown promise, but resulting prediction models await validation [15,16].

The lack of scientific knowledge on treatment moderators is a barrier for efforts to develop evidence-based guidelines to help select the optimal depression treatment for an individual [17]. Current standard practice when selecting treatments for depression is largely guided by practical factors (cost, availability) or factors like clinical judgement and patient preference, which have been shown to be unrelated to symptom response [18]. All mental health stakeholders would benefit from more accurate and more personalized treatment selection strategies [19]. Research to generate precision treatment algorithms can help inform shared decision-making during the treatment selection process [18,20], and is one of many promising avenues for improving outcomes for individuals with depression [21].

In this article, we describe the protocol for a study that aims to improve personalized treatment selection for depression. We overcome the limitations of prior research by developing a model to predict relative treatment effects at the individual patient level. This model will cover five major empirically-supported depression treatments [1,22,23], will be based on multiple person and disorder characteristics, and will be evaluated for its generalizability and clinical performance.

To develop this model, we will conduct a systematic review and individual participant data (IPD) network meta-analysis (NMA) [24] comparing ADM, CT, BA, IPT, and STPP for adult depression on depressive symptom measures at treatment completion, including various available participant characteristics as potential effect modifiers. Estimation of effect modifiers becomes more reliable by combining data from multiple trials, and by focusing on IPD rather than on published aggregate data [24,25]. Furthermore, by adopting a network-based approach instead of focusing on pairwise comparisons, we can estimate treatment effects for multiple treatment comparisons (rather than only two). Moreover, IPD-NMA adequately accounts for the clustering of data within the studies and offers increased statistical power by incorporating both direct and indirect evidence [26–28]. We next describe the protocol for this study, outlining the methods that we will follow.

## 2. Methods

### 2.1. Design and pre-registration

This protocol builds on and extends prior work by our group [11,15,29–31]. This IPD-NMA will be pre-registered in the PROSPERO International prospective register of systematic reviews after acceptance of this protocol for publication in a peer-reviewed journal [29–31]. Additional important protocol amendments will be updated in this register [29–31]. Extraction of data from the primary datasets into the database for this project will start after PROSPERO registration. The Preferred Reporting Items for Systematic Review and Meta-Analysis Protocols (PRISMA-P) [32] and the Transparent Reporting of Multivariable Prediction Models Developed or Validated using Clustered Data (TRIPOD-Cluster) statements [33] guide the writing of this article (see S1 and S2 Tables), and a detailed analytic plan for predictive modelling will be pre-registered in the Open Science Framework.

### 2.2. Eligibility criteria

Eligible studies are randomized clinical trials for adult acute-phase depression comparing two or more of the following interventions: ADM, CT, BA, IPT, STPP. No restrictions will be placed concerning the years when the study was conducted, or with regard to publication language, date, or status [29–31].

 **2.2.1. Participants.** Participants will be considered depressed if they meet specified criteria (e.g., Diagnostic and Statistical Manual of Mental Disorders) for major depressive disorder or another unipolar mood disorder assessed by means of a semi-structured interview or clinicians' assessment, or if they present a score at or above a validated cut-off indicating the likelihood of clinically significant depressive symptoms on an evaluator-assessed, clinician-assessed, or self-reported measure of depression (e.g., Hamilton Depression Rating Scale total score ≥ 10; Beck Depression Inventory total score ≥ 10) [29–31,34]. Comorbid mental and somatic disorders will be allowed [29–31,35]. Participants must be at least 18 years old, and we will place no restrictions concerning the maximum age of study participants [29–31,36].

 We will assess eligibility criteria at study level rather than on the individual participant level [29–31]. Eligible participants from a study including a wider population (e.g., adults from a

study including both adolescents and adults) will not be included, because the integrity of randomization within the subgroup of eligible participants could be compromised [29–31]. We will include all participants randomized to treatment, and adopt an intention-to-treat analysis [29–31].

**2.2.2. Interventions.** We will focus on five major empirically-supported acute-phase depression treatments for which we expect enough trials including head-to-head comparisons with the other treatments are available to obtain relatively precise effect estimates. These five treatments are summarized below (for more extensive definitions we refer elsewhere [1]). We will not include control conditions because their inclusion may change the nature of comparisons and hence the relative effect sizes among the treatment conditions [37], whereas the comparisons of clinical interest are among the alternative treatments.

Concerning ADM, we will consider any type of standard oral antidepressant within the therapeutic dose range (e.g., selective serotonin [and noradrenaline] reuptake inhibitors, tricyclic antidepressants, monoamine oxidase inhibitors) [30].

CT aims to correct maladaptive thinking and beliefs to reduce depressive symptoms [31,38]. BA aims to increase a person's access to positively reinforcing stimuli [31,39,40]. A core technique in BA is activity scheduling, whereby individuals monitor their mood and daily activities to learn the connection between them, and then focus on increasing activities that are expected to result in a sense of pleasure, mastery, or accomplishment [31,39]. This behavioral intervention is also part of CT for depression [38] and the term cognitive–behavioral therapy (CBT) is often used in the literature to denote a single depression intervention that includes both a cognitive restructuring and a behavioral activation component [31].

We will consider an intervention to be CT if it is a manualized psychotherapy with cognitive restructuring as a main treatment component [11,31]. Behavioural techniques (e.g., activity scheduling) will be allowed, as long as they are part of an intervention protocol that is aimed at cognitive restructuring [31]. Beck's model [38] is considered the CT prototype, but other models are also eligible for inclusion [31]. We will consider an intervention to be BA if the core element of treatment is aimed at increasing positive reinforcement by means of activity scheduling [31]. Inclusion of cognitive restructuring techniques in BA will not be allowed [31].

IPT is a structured, time-limited intervention specifically developed for the treatment of major depression that focuses on current salient relational and interpersonal experiences [30]. We will consider an intervention to be IPT when it is a psychotherapy based on the manuals developed by Klerman and Weissman for IPT or for the briefer version called interpersonal counselling [30,41–45].

STPP is rooted in psychoanalytic theories, which consider the underlying personality structure to play an important role in the development and maintenance of depression. STPP aims to foster insight into (past) interpersonal relationships and unconscious feelings, desires, strivings, and thoughts to treat depression. We will consider an intervention to be STPP if it is based on psychoanalytic theories and practices, is time-limited from the onset (i.e., not a therapy that is brief only in retrospect) to distinguish it from long-term psychodynamic psychotherapy, and applies verbal techniques (e.g., therapies applying art as expression form are not considered STPP) [29].

We will include psychotherapies in any delivery format (i.e., face-to-face, telephone, or videoconferencing), as long as a clinician delivers the therapy [30,31]. Bibliotherapy, internet therapy, or other self-help formats will be excluded, as will be blended treatment formats that combine clinician-delivered therapy with internet interventions. Treatment must not exceed 6 months with no restrictions on the number of sessions [30,31]. Inpatient settings, partial hospitalization programs, and intensive outpatient programs will be excluded, since by definition more

care is provided than psychotherapy or antidepressant monotherapy. We will place no other restrictions on the setting in which treatment is delivered (e.g., primary care, outpatient mental health care) [30,31]. We will only include studies of acute-phase depression treatment [30,31]. Thus, we will exclude, for example, studies that randomized participants to the interventions as maintenance treatments after successful acute treatment [30,31]. Only data from the relevant conditions and comparisons of eligible studies will be included [30]. For example, for studies involving augmentation of antidepressants or psychotherapy following non-response to mono-therapy, only study data up until the first triage point for augmentation will be included [30,46]. Similarly, for studies involving other conditions (e.g., [placebo] control, combined treatment of ADM and psychotherapy), data from these conditions will not be included.

## 2.3. Outcome

Depressive symptom level at treatment completion will be the primary outcome of this study, as symptom reduction is considered to be the main aim of acute-phase depression treatments [29,31]. We chose a continuous measure over categorical outcomes (e.g., remission) as our primary target because dichotomizing continuous variables reduces statistical power for esti-mating interactions with treatment [25,29–31].

Depressive symptom level at treatment completion is operationalized as a participant's score on the Beck Depression Inventory-II (BDI-II) [47] at the primary post-treatment time point as defined by the study's authors [30,31]. We chose the BDI-II as the main outcome because it is a self-reported patient-centered instrument that was designed to measure depres-sion severity [47], is frequently applied in depression treatment research, and is sensitive to change [48]. If collected study data does not include BDI-II, we will convert scores on the primary (secondary, third, fourth, etc. in this order) continuous depression scale, as defined by the study's authors, into BDI-II scores using existing conversion algorithms [49–51]. If none of the measures assessed in the study is included in conversion algorithms, we aim to develop one ourselves based on the collected IPD. When none of the used outcome scales can be justifiably converted into the BDI-II, we will exclude the study from analysis. If this results in three or more excluded studies, we will consider transforming primary continuous depres-sion outcomes to a 0–100 scale for all studies.

## 2.4. Predictors

Moderators of treatment effect (elsewhere also referred to as "prescriptive predictors", "effect modifiers" or "treatment-covariate interactions") affect the direction or magnitude of differ-ences in outcome between two treatments [52], and thus can help predict whether a patient will benefit more from one treatment than another [18]. Variables that moderate individuals' response to treatment can also have prognostic (or "main") effects (see Cohen & DeRubeis [18] for additional discussion), which will also be included in the model.

Putative predictor variables include demographic (e.g., age, gender), clinical (e.g., depres-sion severity, depressive episode duration, comorbid anxiety disorder), and psychological (e.g., personality, coping style) participant characteristics assessed before the start of treatment [29–31]. Candidate variables will be selected for inclusion in the model based on a recent systematic review on predictors of treatment response in depression and sufficient availability across the studies, following the approach described by Vale and colleagues [53].

## 2.5. Systematic literature search

To identify eligible studies, we will use a search strategy that has been described previously [30,31]. We will search a database of randomized clinical trials examining the efficacy and

effectiveness of psychological treatments for depression that has been used in a series of published meta-analyses (www.metapsy.org) [54]. This database was developed through comprehensive literature searches in the bibliographic databases PubMed, PsycINFO, Embase.com, and the Cochrane Library and is updated every four months. The search strings use a combination of index terms and free-text words indicative of depression and psychotherapies, with filters for randomized clinical trials. The exact search terms can be found at https://osf.io/nv3ea/. Two raters independently screen all records, assess full-text papers for METAPSY database eligibility, and categorize the treatment comparison(s) of included studies, with disagreements being resolved in consensus in each phase [30,31].

Two other raters will independently assess all full-text papers of studies marked as comparing a psychotherapy mono-treatment condition against another active (i.e., psychotherapy or ADM) mono-treatment condition for meeting the inclusion criteria for this work [30,31]. Disagreements will be resolved through consensus and if consensus cannot be reached, a third rater will be consulted [29–31]. To identify studies that might have been missed, we will check the references of prior reviews and meta-analyses [30,31]. We will also contact psychotherapy listservs to request ongoing or unpublished studies, and studies that were missed [29,30].

## 2.6. IPD collection

We will invite authors of eligible studies to participate in this project using a strategy that has been successful in soliciting participation in previous depression treatment IPD meta-analyses [11,29–31]. We will invite representatives of each team that shares IPD to join as authors on all publications resulting from the use of these data, inasmuch as they meet internationally accepted criteria for authoring scientific articles (www.icmje.org) [29–31]. In addition, we will make the combined database available to investigators sharing IPD to examine other research questions, provided that the authors of the original studies approve the use of their data for this purpose [11,29–31,55].

We will apply a multi-step protocol to contact authors. We have described this protocol in more detail previously [29–31] and it has proven to be successful in reaching authors for depression treatment IPD meta-analyses. In short, we will contact the authors (starting with the corresponding author and proceeding to the other co-authors) by email with three reminders. In case of no response to email, we will first mail a letter to the corresponding author (again with three attempts) and then contact the corresponding author by telephone. This is followed by contacting the other authors by letter and telephone, and other ways to contact one of the authors (e.g., via colleagues who might know them) [29–31]. A study's data will be considered unavailable only if all these attempts fail, or an author either indicates that the IPD were not retained or declines to share these data [29–31].

We will request the following anonymized participant-level data items: randomized treatment condition, all outcome variables assessed prior to, during, and after treatment (with item-level data for depression outcome measures), and all potential predictor variables assessed in the study [30,31]. We will also ask for the study protocol and the assessment batteries used in the study.

The following study-level characteristics are extracted from the publication by two independent raters upon inclusion in the METAPSY database: country, recruitment method (e.g., community, clinical), target group (e.g., adults in general, students), depression inclusion criteria, number of treatment sessions, treatment format (e.g., individual, group), and assessment time points [30,31]. We will examine whether the ADM treatment protocols meet minimum adequate clinical best practice guidelines by assessing studies for the use of a therapeutic dosage and titration schedule (i.e., therapeutic dose achieved within three weeks) [30]. Study ADM will be deemed adequate if both criteria are met [27,30]. We will examine psychotherapy

protocol adequacy with regard to use of a treatment manual, provision of therapy by trained therapists, and verification of treatment integrity [29–31]. Treatment quality characteristics will be extracted by two raters independently and disagreements will be resolved by consensus [29–31]. If consensus cannot be reached, a third rater will be consulted. Characteristics not reported in the publications will be requested from the authors [29–31].

Risk of bias in the included studies will be assessed using the Cochrane risk-of-bias tool for randomized trials version 2 [56], both before and after the IPD are obtained. Ratings will be primarily based on information reported in the publications, though risk of bias due to deviations from the intended intervention, due to missing outcome data, and in selection of the reported result will be assessed using the IPD (i.e., domain 3.1) or the IPD meta-analysis' abilities to correct for the relevant risk of bias (i.e., domains 2.6–2.7, 3.2, and 5.1–5.3). Two raters will independently assess the risk-of-bias tool at study level. Disagreements will be resolved by consensus [30,31]. If consensus cannot be reached a third rater will be consulted. As studies are expected to be included that were published before the universal adoption of reporting guidelines for randomized clinical trials [57], requisite information will be requested from the authors if not reported in the publications [31].

### 2.7. IPD integrity checks

We will perform three data integrity checks [29–31]. First, we will check whether the dataset includes the full intention-to-treat sample (defined as all participants randomized to treatment in the study) and otherwise matches the data reported in the published article [29–31]. To this end, means and standard deviations or raw counts for all baseline characteristics, and observed pre-treatment and post-treatment scores reported in the article will also be calculated from the dataset and both will be compared [29–31]. Second, we will check whether all outcome and potential predictor variables reported in the article are included in the dataset [29–31]. Third, the outcome and predictor variables will be checked for inconsistent, invalid, or out-of-range values [29–31], including values that conflict with the primary study's or this work's eligibility criteria (e.g., age < 18). Discrepancies resulting from these data integrity checks will be discussed with the authors [29–31]. In addition, authors will be contacted if concerns arise regarding duplicate participants across trials (e.g., when multiple trials are conducted by the same author group).

### 2.8. Building the IPD database

All datasets received that contain IPD for at least one continuous depressive symptom measure will be considered for quantitative synthesis and will be pooled in a single database [29–31]. Predictor variables are expected to be assessed differently across individual studies and will therefore be harmonized [29–31]. Categorical variables (e.g., marital status) will be recoded so that they contain similar categories and corresponding coding schemes across all studies that include that variable (e.g., 0=single, 1=married, 2=separated/divorced, 3=widowed, 4=not married, but living together, 5=other).

To improve rigor and reproducibility of our data processing and cleaning pipelines, we developed a spreadsheet- and code-based system that also increases efficiency and scalability, allowing for changes to be instantaneously implemented across the database, and facilitating addition of new data. We will describe this system in more detail on the project's Open Science Framework page.

### 2.9. Missing data imputation

Data will be imputed for variables that are 'systematically' missing for certain studies and values that are 'sporadically' missing for certain individual participants. We will use methods for

multiple imputation that account for the clustering of participants in different studies [58,59], utilizing both baseline variables as well as all intermediate and endpoint outcomes available in the datasets [60]. If there are no systematically missing predictors in the studies, we will perform multiple imputation of missing data for each study separately as our primary analysis, and use the procedure described above as sensitivity analysis. Single imputation will also be considered, for instance for imputation of baseline variables, if it follows recommended advice [61].

## 2.10. Modelling approach

First, we will estimate the average relative treatment effects (mean differences) using pairwise meta-analyses and a random effects NMA based on aggregate data. NMA enables us to obtain relative treatment effects for all treatment comparisons, even for those without direct evidence [62], as long as they are connected in the network. One prerequisite for NMA is transitivity, i.e., that the included studies do not differ with respect to the distribution of effect modifiers when grouped by comparison. Transitivity also means that a 'mega-trial' would be possible, i.e., a hypothetical trial comparing all treatments in the network, and that all patients would in principle be eligible to receive any of these treatments. We will assess transitivity conceptually and by examining candidate effect modifiers across comparisons. Lack of transitivity may affect the consistency of the network, or the extent to which the direct and indirect estimated effect of a particular comparison align. When direct and indirect estimated effects are inconsistent (i.e., do not align), NMA results may be invalid. Therefore, we will evaluate statistical consistency using the design-by-treatment interaction model [63] and the back calculation method [64]. We will assume a common heterogeneity parameter in the network, and we will assess its extent by looking at prediction intervals. For these analyses, we will use the netmeta package in R [65].

To model the relative individualized treatment effects amongst the five treatments, we will conduct a two-stage IPD-NMA in the rjags R package [66]. In the first stage, we will analyze each study separately using a suitable regression model (e.g., linear regression adjusting for baseline) to estimate the treatment effect at average values of the predictors as well as all treatment-predictor interactions. For estimating treatment-predictor interactions, we will consider using penalization (shrinkage). Such methods tend to shrink (decrease the values of) the estimated coefficients as compared to usual maximum likelihood approaches. A simulation has shown that penalization methods may lead to more accurate predictions (i.e., smaller mean squared error) regarding patient-specific treatment effects in IPD meta-analysis compared to unpenalized methods [67]. We will fit the model in each of the multiply imputed datasets for each study. Then, we will combine posterior samples of the model parameters across the imputed datasets, to estimate study-level parameters and their corresponding uncertainty [68].

In the second stage, we will synthesize all study-specific results (treatment effect estimates, and interactions, and their standard errors) using a (multivariate) NMA model, where, as above, we will assume common between-study heterogeneity across all treatment comparisons. The end product will be a set of estimates of the treatment effect at average values of the predictors as well as treatment-predictor interactions (i.e., effect modification). The final model will allow us, given a participant's baseline values, to predict individualized treatment effects between any two treatments ($T_A$ and $T_B$) in the network, where treatment effect means the predicted difference in outcome if the individual received $T_A$ versus $T_B$.

## 2.11. Cross-validation of the model

We will assess the performance of the model with internal-external cross-validation [69,70]. More specifically, we will repeat the analytical process described above after taking one

study out of our sample and use the remaining studies to fit our model. Then, we will apply the model to the left-out study, and for each patient therein we will calculate the predicted treatment effects among all treatments in the study. Finally, using these predictions, we will evaluate model performance within and across studies using meta-analysis methods. Conceptually, this evaluation will follow the Personalized Advantage Index approach described by DeRubeis and colleagues [71] in which outcomes of participants who received the treatment recommended by the model are compared against the outcomes of participants who received a non-recommended treatment [71]. This has been also termed "population benefit" index [72]. This measure combines aspects of discrimination-for-benefit and calibration-for-benefit. A useful model for personalized treatment effects should have an index larger than the average treatment effect. We will also assess calibration-for-benefit purely by producing calibration plots and fitting a regression for benefit [72]. Analyses will be done with the predieval R package (v.0.1.1) [73].

## 2.12. Assessment of meta-biases

To assess the potential impact of risk-of-bias, we will conduct a random effect NMA excluding studies at high risk of bias. We will visually compare the 95% confidence intervals of the mean relative treatment effects for all treatment comparisons between this and the primary analysis.

We will assess potential data-availability bias by comparing effect sizes (i.e., mean relative treatment effects) between studies for which IPD were and were not available with a subgroup NMA. To facilitate comparability, effect sizes will be calculated based on the effect size data extracted from the publications upon inclusion in the METAPSY database for all studies (rather than being based on the IPD when available) [31].

Small study effects (potential publication bias) will be assessed by examining asymmetry in contour-enhanced funnel plots with Egger's test of the intercept for pairwise comparisons including ten or more trials [30,31,74]. The analyses will include both studies for which IPD were versus were not available and will, therefore, also be based on the effect size data extracted from the publications [31]. We will assess the strength of the body of evidence based on the number of included studies and participants as well as the quality of the included studies [30,31].

## 2.13. Ethical standards

Institutional Review Board (IRB) approval is not required for this project, because we will work with anonymized data from treatment studies that have already been completed [29–31]. IRB approval might be required for the investigators to share their IPD depending on their institution's policies [29–31]. It will be the responsibility of the investigators to obtain IRB approval if their institution's policies require them to do so [29–31]. By signing the data sharing agreement, the authors who share their IPD will declare that those data were collected and will be transferred according to all applicable local and international laws and regulations [29–31]. Furthermore, they declare that all IPD will be anonymized, so that no personal data are transferred [31].

## 3. Discussion

In this article, we described the rationale and protocol for a study that aims to improve personalized treatment selection for depression. More specifically, we aim to build an IPD database from existing randomized clinical trials worldwide covering five major empirically-supported depression treatments, and to conduct an IPD-NMA to estimate individualized relative treatment effects for these five interventions.

This study has several strengths. First, we describe a data-analysis strategy that combines state-of-the-art methods to predict personalized treatment effects based on combined data from multiple clinical trials. Second, we will build a unique patient-level database of worldwide trials covering five of the most widely used depression treatments. We will make this database available for future research by ourselves and others, beyond the scope and duration of this project, thus prospectively reducing the burden that clinical research poses on patients. Moreover, we will build the lasting infrastructure that will allow this database to further expand (e.g., adding new studies), as well as systems that could be used to develop similar IPD databases in other contexts. Third, applying the data-analysis strategy in this database will result in a prediction model, which is based on multiple observed person and illness characteristics and results in directly usable clinical recommendations for optimizing treatment selection for individuals in the context of multiple treatments. Overcoming an important limitation of prior research in this field, we will evaluate this model and assess its potential generalizability. We will examine model performance using internal-external cross-validation to reduce the potential for biased estimates and overfitting.

This study also has several limitations. First, primary studies will likely show considerable heterogeneity in key features such as which and how predictors were assessed, number and frequency of treatment sessions, and assessment timing. Harmonization of variables across relevant studies will also be necessary to account for variation in how predictors were operationalized across different studies [29–31]. Recoding categorical variables into a smaller number of categories might be necessary, despite the potential resulting loss of information [29–31]. Second, this study does not cover all available depression treatments. We have focused on five widely used and studied interventions for which we expect a substantial amount of comparative IPD will be available. However, our methodology will be adaptable for future inclusion of low-intensity treatments (e.g., psychoeducation), other psychotherapies (e.g., problem-solving therapy), combined treatment of psychotherapy and ADM [46], or depression treatments in different contexts (e.g., digital therapy, inpatient treatment, or brain stimulation therapies). Third, although identification of studies is based on systematic literature searches, it is possible that we might miss studies. For instance, the searches underlying the METAPSY database do not include a formal grey literature search. In addition, we expect that IPD will not be obtained for all studies that we do identify. Although a recent study found few indications for data-availability bias in IPD meta-analyses [75], we will assess empirically the extent to which mean relative treatment effects differ between studies for which IPD are versus are not available. We also will examine empirically potential publication bias.

If accuracy and generalizability of the resulting statistical model's predictions are deemed adequate, it could be used to develop a depression treatment selection tool. This tool would need to be evaluated prospectively in clinical practice before it can be implemented in mental health care to facilitate knowledge-based decision making, for instance by prospective randomized comparisons of model-informed treatment allocation versus allocation-as-usual or randomized allocation [76]. Future work should also focus on developing and evaluating implementation strategies for precision mental health approaches [19]. We hope that the results of this study will be evaluated further for their potential to inform shared decision making and help depressed individuals receive the optimal treatment given their personal characteristics. In this way, we hope to contribute to personalizing evidence-based treatment selection for depression to reduce this disorder's tremendous personal and societal costs.

## Supporting information

**S1 Table.** PRISMA-P (Preferred Reporting Items for Systematic Review and Meta-Analysis Protocols) 2015 Checklist: Recommended Items to Address in a Systematic Review Protocol. (DOCX)

**S2 Table.** TRIPOD-Cluster Checklist of Items to Include When Reporting a Study Developing or Validating a Multivariable Prediction Model Using Clustered Data.
(DOCX)

## Author contributions

**Conceptualization:** Ellen Driessen, Pim Cuijpers, Marjolein Fokkema, Steven D. Hollon, Jos W. R. Twisk, Zachary D. Cohen.

**Data curation:** Zachary D. Cohen.

**Funding acquisition:** Ellen Driessen, Pim Cuijpers, Marjolein Fokkema, Steven D. Hollon, Jos W. R. Twisk, Zachary D. Cohen.

**Investigation:** Ellen Driessen, Frederik J. Wienicke, Jasmijn Breunese, Pim Cuijpers, Steven D. Hollon, Anuj H. P. Mehta, Madison R. Schmidt, Zachary D. Cohen.

**Methodology:** Ellen Driessen, Orestis Efthimiou, Frederik J. Wienicke, Pim Cuijpers, Thomas P. A. Debray, David J. Fisher, Marjolein Fokkema, Toshiaki A. Furukawa, Steven D. Hollon, Richard D. Riley, Jos W. R. Twisk, Zachary D. Cohen.

**Project administration:** Ellen Driessen, Jasmijn Breunese, Madison R. Schmidt.

**Software:** Zachary D. Cohen.

**Supervision:** Ellen Driessen, Zachary D. Cohen.

**Writing – original draft:** Ellen Driessen, Orestis Efthimiou, Frederik J. Wienicke, Jasmijn Breunese, Zachary D. Cohen.

**Writing – review & editing:** Ellen Driessen, Orestis Efthimiou, Frederik J. Wienicke, Jasmijn Breunese, Pim Cuijpers, Thomas P. A. Debray, David J. Fisher, Marjolein Fokkema, Toshiaki A. Furukawa, Steven D. Hollon, Anuj H. P. Mehta, Richard D. Riley, Madison R. Schmidt, Jos W. R. Twisk, Zachary D. Cohen.

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
