## [Decision Letter · Decision Letter 0]

27 Sep 2024

PONE-D-23-39716Developing a multivariable prediction model to support personalized selection among five major empirically-supported treatments for adult depression. Study protocol of a systematic review and individual participant data network meta-analysis.PLOS ONE

Dear Dr. Cohen,

Thank you for submitting your manuscript to PLOS ONE. After careful consideration, we feel that it has merit but does not fully meet PLOS ONE’s publication criteria as it currently stands. Therefore, we invite you to submit a revised version of the manuscript that addresses the points raised during the review process.

We look forward to receiving your revised manuscript.

Kind regards,

Nishant Premnath Jaiswal, MBBS, PhD

Academic Editor

PLOS ONE

https://eprints.whiterose.ac.uk/190928/1/efficacy-and-moderators-of-cognitive-therapy-versus-behavioural-activation-for-adults-with-depression-study-protocol-of-a-systematic-review-and-meta-analysis-of-individual-participant-data.pdf?

In your revision ensure you cite all your sources (including your own works), and quote or rephrase any duplicated text outside the methods section. Further consideration is dependent on these concerns being addressed.

3. In the online submission form, you indicated that the collective de-identified individual participant database that will be developed for this study will be available for use by other researchers, provided that the authors of the original studies approve the use of their data for this purpose. Requests can be made with the corresponding author (ellen.driessen@ru.nl). Access (with limited investigator support) will be granted after approval of a study proposal by all authors and a signed data access agreement. 

4. As required by our policy on Data Availability, please ensure your manuscript or supplementary information includes the following: 

Reviewers' comments:

Reviewer's Responses to Questions

**Comments to the Author**

1. Does the manuscript provide a valid rationale for the proposed study, with clearly identified and justified research questions?

Reviewer #1: Yes

2. Is the protocol technically sound and planned in a manner that will lead to a meaningful outcome and allow testing the stated hypotheses?

Reviewer #1: Yes

3. Is the methodology feasible and described in sufficient detail to allow the work to be replicable?

Reviewer #1: Yes

4. Have the authors described where all data underlying the findings will be made available when the study is complete?

Reviewer #1: Yes

5. Is the manuscript presented in an intelligible fashion and written in standard English?

Reviewer #1: Yes

6. Review Comments to the Author

You may also provide optional suggestions and comments to authors that they might find helpful in planning their study.

Reviewer #1: Thank you for the opportunity to review this protocol for an outstanding study. This is one of the most well-written protocols and most promising trials I have recently seen. I have only a few minor comments.

1. There seems to be some confusion regarding who is the corresponding author of this manuscript (ED or ZDC).

2. The authors may want to include the following reference concerning treatment heterogeneity in the introduction section:

Volkmann C, Volkmann A, Müller CA. On the treatment effect heterogeneity of antidepressants in major depression: A Bayesian meta-analysis and simulation study. Hutson AD, ed. PLoS One. 2020;15(11):e0241497. doi:10.1371/journal.pone.0241497

3. I have a question regarding the included delivery formats. The manuscript states that guided digital interventions will be included. Are there any restrictions regarding the amount of guidance? This can differ significantly, to the point where human support is reduced to a couple of minutes per patient.

4. How will the authors account for the effect of different settings/intensities of care, as both outpatient and inpatient care settings are included?

5. Will there be a minimum depression baseline score for inclusion? How will this be operationalized?

6. Will there be exclusions of participants on an individual patient data level if the authors find indications that some inclusion criteria are not met by single persons in the IPD?

7. Regarding data checks, it appears that the authors do not compare the means/standard deviations or raw counts with those reported in the publication. This is highly recommended.

8. Multiple imputation of missing moderators (even if they are entirely missing in a study): I am not sufficiently familiar with Network Meta-Analyses to comment in detail, but if feasible, I recommend performing a sensitivity analysis where imputation is conducted within each study only.

7. PLOS authors have the option to publish the peer review history of their article (what does this mean? ). If published, this will include your full peer review and any attached files.

**Do you want your identity to be public for this peer review?** For information about this choice, including consent withdrawal, please see our Privacy Policy .

Reviewer #1: No

---

## [Author Response · Author response to Decision Letter 1]

7 Nov 2024

Response:

We have double checked the manuscript against the abovementioned templates to ensure it meets PLOS ONE’s style requirements. As a result, we have removed abbreviations in the author affiliations (spelling out full terms instead), adapted heading formats, and updated file names.

https://eprints.whiterose.ac.uk/190928/1/efficacy-and-moderators-of-cognitive-therapy-versus-behavioural-activation-for-adults-with-depression-study-protocol-of-a-systematic-review-and-meta-analysis-of-individual-participant-data.pdf?

In your revision ensure you cite all your sources (including your own works), and quote or rephrase any duplicated text outside the methods section. Further consideration is dependent on these concerns being addressed.

Response:

We have compared the manuscript with the abovementioned previous publication by our group. As a consequence, we have added citations in the methods section and rephrased three sentences in the discussion section. In this way, we hope to have addressed all minor text overlap. If, despite our best efforts, any occurrences can still be identified, please let us know which specific sections are of concern and we will be more than happy to address these too.

3. In the online submission form, you indicated that the collective de-identified individual participant database that will be developed for this study will be available for use by other researchers, provided that the authors of the original studies approve the use of their data for this purpose. Requests can be made with the corresponding author (ellen.driessen@ru.nl). Access (with limited investigator support) will be granted after approval of a study proposal by all authors and a signed data access agreement.

Response:

This manuscript describes the protocol for a study that has not yet been conducted. Therefore, it does not describe any findings. The individual participant database described above has not yet been developed and it is currently not possible to make this available.

We can see how our phrasing in the online submission form might be confusing in this regard and have now adjusted it as follows:

Data availability is not applicable to this article as no data were analyzed for this study protocol paper.

4. As required by our policy on Data Availability, please ensure your manuscript or supplementary information includes the following:

Response:

This manuscript is a study protocol paper and does not describe any findings. Literature searches, data extraction, and risk of bias assessments have not yet been completed. These will be described in the article reporting the outcomes of the study we describe in the current protocol paper.

Response:

We now include captions for the Supporting Information files at the end of the manuscript and have updated their in-text citations accordingly.

Reviewers' comments:

Reviewer's Responses to Questions

Comments to the Author

1. Does the manuscript provide a valid rationale for the proposed study, with clearly identified and justified research questions?

Reviewer #1: Yes

2. Is the protocol technically sound and planned in a manner that will lead to a meaningful outcome and allow testing the stated hypotheses?

Reviewer #1: Yes

3. Is the methodology feasible and described in sufficient detail to allow the work to be replicable?

Reviewer #1: Yes

4. Have the authors described where all data underlying the findings will be made available when the study is complete?

Reviewer #1: Yes

5. Is the manuscript presented in an intelligible fashion and written in standard English?

Reviewer #1: Yes

6. Review Comments to the Author

You may also provide optional suggestions and comments to authors that they might find helpful in planning their study.

Reviewer #1: Thank you for the opportunity to review this protocol for an outstanding study. This is one of the most well-written protocols and most promising trials I have recently seen. I have only a few minor comments.

Response:

We thank Reviewer 1 for their thoughtful review and positive feedback. We are very happy to hear that our proposed study is considered promising and of high quality.

1. There seems to be some confusion regarding who is the corresponding author of this manuscript (ED or ZDC).

Response:

ED and ZDC share corresponding authorship for this manuscript and are, therefore, both listed as corresponding authors. This is in line with PLOS ONE’s submission guidelines that do “not restrict the number of corresponding authors that may be listed on the article in the event of publication” (https://journals.plos.org/plosone/s/submission-guidelines).

2. The authors may want to include the following reference concerning treatment heterogeneity in the introduction section:

Volkmann C, Volkmann A, Müller CA. On the treatment effect heterogeneity of antidepressants in major depression: A Bayesian meta-analysis and simulation study. Hutson AD, ed. PLoS One. 2020;15(11):e0241497. doi:10.1371/journal.pone.0241497

Response:

We thank Reviewer 1 for this suggestion and have included the reference [5] in the introduction (page 5, marked up copy) as follows:

People suffering from depression have a range of therapeutic options, including various antidepressant medications (ADMs) and psychological treatments such as cognitive therapy (CT), behavioral activation (BA), interpersonal psychotherapy (IPT), and short-term psychodynamic psychotherapy (STPP). Although no significant differences in average treatment effects have been found between these interventions [1,2], response is highly heterogeneous [3,4,5] warranting the need for more personalized treatment recommendations [6].

3. I have a question regarding the included delivery formats. The manuscript states that guided digital interventions will be included. Are there any restrictions regarding the amount of guidance? This can differ significantly, to the point where human support is reduced to a couple of minutes per patient.

Response:

This is a good point. To avoid large heterogeneity in the amount of guidance between studies, we decided to exclude guided digital interventions. Our literature searches so far did not identify any study of guided digital interventions meeting the inclusion criteria for this work, suggesting that this decision likely will not lead to a substantial decrease in sample size. We have adjusted the relevant methods section (page 9, marked up copy), which now reads:

We will include psychotherapies in any delivery format (i.e., face-to-face, telephone, or videoconferencing), as long as a clinician delivers the therapy [30,31]. Bibliotherapy, internet therapy, or other self-help formats will be excluded.

4. How will the authors account for the effect of different settings/intensities of care, as both outpatient and inpatient care settings are included?

Response:

We thank Reviewer 1 for this question. We realize now that inpatient psychiatric care settings by definition do not meet our inclusion criteria, as do partial hospitalization and intensive outpatient programs, since patients in these settings receive more care than psychotherapy or antidepressant monotherapy. Therefore, we have rephrased the relevant section as follows to avoid this confusion (page 9, marked up copy):

Treatment must not exceed 6 months with no restrictions on the number of sessions [30,31]. Inpatient settings, partial hospitalization programs, and intensive outpatient programs will be excluded, since by definition more care is provided than psychotherapy or antidepressant monotherapy. We will place no other restrictions on the setting in which treatment is delivered (e.g., primary care, outpatient mental health care) [30,31].

5. Will there be a minimum depression baseline score for inclusion? How will this be operationalized?

Response:

As described at page 7 (marked up copy), depression is defined at study-level, as participants 1) meeting specified criteria (e.g., Diagnostic and Statistical Manual of Mental Disorders) for major depressive disorder or another unipolar mood disorder assessed by means of a semi-structured interview or clinicians’ assessment, or 2) presenting a score at or above a validated cut-off indicating the likelihood of clinically significant depressive symptoms on an evaluator-assessed, clinician-assessed, or self-reported measure of depression.

As such, there is no minimum depression baseline score for inclusion. For studies in the first category, baseline depression scores are allowed to vary as long as the participants meet diagnostic criteria for a unipolar mood disorder. For studies in the second category, the minimum depression baseline score is operationalized by the cut-off score of the specific measure used indicating the likelihood of clinically significant depressive symptoms (e.g., Hamilton Depression Rating Scale score ≥ 10; Beck Depression Inventory score ≥ 10).

We have now rephased the relevant section in hopes of providing further clarification:

Participants will be considered depressed if they meet specified criteria (e.g., Diagnostic and Statistical Manual of Mental Disorders) for major depressive disorder or another unipolar mood disorder assessed by means of a semi-structured interview or clinicians’ assessment, or if they present a score at or above a validated cut-off indicating the likelihood of clinically significant depressive symptoms on an evaluator-assessed, clinician-assessed, or self-reported measure of depression (e.g., Hamilton Depression Rating Scale total score ≥ 10; Beck Depression Inventory total score ≥ 10) [29-31,34].

6. Will there be exclusions of participants on an individual patient data level if the authors find indications that some inclusion criteria are not met by single persons in the IPD?

Response:

Yes, it is part of our data integrity check procedure to check for participants reporting values that are in conflict with the eligibility criteria of the study in which they were enrolled or the meta-analysis proposed (e.g., age <18). We considered this included in our checks “for inconsistent, invalid, or out-of-range values”, but we now describe this explicitly in the relevant methods section (page 14, marked up copy) as follows:

Third, the outcome and predictor variables will be checked for inconsistent, invalid, or out-of-range values [29-31], including values that conflict with the primary study’s or this work’s eligibility criteria (e.g., ag

---

## [Decision Letter · Decision Letter 1]

4 Feb 2025

PONE-D-23-39716R1Developing a multivariable prediction model to support personalized selection among five major empirically-supported treatments for adult depression. Study protocol of a systematic review and individual participant data network meta-analysis.PLOS ONE

Dear Dr. Cohen,

Thank you for submitting your manuscript to PLOS ONE. After careful consideration, we feel that it has merit but does not fully meet PLOS ONE’s publication criteria as it currently stands. Therefore, we invite you to submit a revised version of the manuscript that addresses the points raised during the review process.

**ACADEMIC EDITOR: **The protocol is well written and explains the methodology and plans in detailsIt will be interesting to see the full review when completed

We look forward to receiving your revised manuscript.

Kind regards,

Nishant Premnath Jaiswal, MBBS, PhD

Academic Editor

PLOS ONE

Journal Requirements:

Reviewers' comments:

Reviewer's Responses to Questions

**Comments to the Author**

1. Does the manuscript provide a valid rationale for the proposed study, with clearly identified and justified research questions?

Reviewer #1: Yes

2. Is the protocol technically sound and planned in a manner that will lead to a meaningful outcome and allow testing the stated hypotheses?

Reviewer #1: Yes

3. Is the methodology feasible and described in sufficient detail to allow the work to be replicable?

Reviewer #1: Yes

4. Have the authors described where all data underlying the findings will be made available when the study is complete?

Reviewer #1: Yes

5. Is the manuscript presented in an intelligible fashion and written in standard English?

Reviewer #1: Yes

6. Review Comments to the Author

You may also provide optional suggestions and comments to authors that they might find helpful in planning their study.

Reviewer #1: Thank you for addressing my comments. Most of my concerns have been thoroughly addressed.

However, there is one detail that may require further clarification:

The authors state: "We will include psychotherapies in any delivery format (i.e., face-to-face, telephone, or videoconferencing), as long as a clinician delivers the therapy [30,31]. Bibliotherapy, internet therapy, or other self-help formats will be excluded."

It is somewhat disappointing that, under the current definition, guided and unguided digital interventions are excluded. These are precisely the types of interventions that hold the greatest potential for scaling psychotherapy treatments in the future.

Nevertheless, the authors might want to consider adding a specification to the inclusion/exclusion criteria clarifying how they plan to deal with mixed-methods interventions, including blended treatment formats.

Thank you once again for the opportunity to review this excellent protocol. I am eagerly looking forward to the results of this study.

7. PLOS authors have the option to publish the peer review history of their article (what does this mean? ). If published, this will include your full peer review and any attached files.

**Do you want your identity to be public for this peer review?** For information about this choice, including consent withdrawal, please see our Privacy Policy .

Reviewer #1: No

---

## [Author Response · Author response to Decision Letter 2]

4 Mar 2025

Journal Requirements:

Reply

We have double-checked the reference list and can ensure it is complete and correct. We do not cite any papers that have been retracted.

Reviewers' comments:

Reviewer #1: Thank you for addressing my comments. Most of my concerns have been thoroughly addressed.

However, there is one detail that may require further clarification:

The authors state: "We will include psychotherapies in any delivery format (i.e., face-to-face, telephone, or videoconferencing), as long as a clinician delivers the therapy [30,31]. Bibliotherapy, internet therapy, or other self-help formats will be excluded."

It is somewhat disappointing that, under the current definition, guided and unguided digital interventions are excluded. These are precisely the types of interventions that hold the greatest potential for scaling psychotherapy treatments in the future.

Nevertheless, the authors might want to consider adding a specification to the inclusion/exclusion criteria clarifying how they plan to deal with mixed-methods interventions, including blended treatment formats.

Thank you once again for the opportunity to review this excellent protocol. I am eagerly looking forward to the results of this study.

Reply

We thank Reviewer 1 for reviewing the revised version of this manuscript. We are very happy to hear that Reviewer 1 considers this protocol of high quality and prior concerns generally addressed.

We acknowledge the potential of digital interventions as scalable depression treatments. Prior work has been conducted aimed at personalizing treatment selection in this area (e.g., https://pubmed.ncbi.nlm.nih.gov/33471111/).

We agree that the manuscript would benefit from specifying how mixed-methods interventions and blended treatment formats are dealt with. We propose to exclude such treatment formats too to avoid large heterogeneity in the amount of guidance between studies. Please note that our literature searches so far did not identify any study applying a blended format. As such, we expect this decision will not lead to many studies being excluded from consideration. We have adjusted the relevant methods section (page 9, marked up copy), which now reads:

We will include psychotherapies in any delivery format (i.e., face-to-face, telephone, or videoconferencing), as long as a clinician delivers the therapy [30,31]. Bibliotherapy, internet therapy, or other self-help formats will be excluded, as will be blended treatment formats that combine clinician-delivered therapy with internet interventions.

---

## [Decision Letter · Decision Letter 2]

18 Mar 2025

Developing a multivariable prediction model to support personalized selection among five major empirically-supported treatments for adult depression. Study protocol of a systematic review and individual participant data network meta-analysis.

PONE-D-23-39716R2

Dear Dr. Cohen,

We’re pleased to inform you that your manuscript has been judged scientifically suitable for publication and will be formally accepted for publication once it meets all outstanding technical requirements.

Kind regards,

Nishant Premnath Jaiswal, MBBS, PhD

Academic Editor

PLOS ONE

Additional Editor Comments (optional):

All the comments have been addressed satisfactorily

Reviewers' comments:

Reviewer's Responses to Questions

**Comments to the Author**

1. Does the manuscript provide a valid rationale for the proposed study, with clearly identified and justified research questions?

Reviewer #1: Yes

2. Is the protocol technically sound and planned in a manner that will lead to a meaningful outcome and allow testing the stated hypotheses?

Reviewer #1: Yes

3. Is the methodology feasible and described in sufficient detail to allow the work to be replicable?

Reviewer #1: Yes

4. Have the authors described where all data underlying the findings will be made available when the study is complete?

Reviewer #1: Yes

5. Is the manuscript presented in an intelligible fashion and written in standard English?

Reviewer #1: Yes

6. Review Comments to the Author

You may also provide optional suggestions and comments to authors that they might find helpful in planning their study.

Reviewer #1: All my questions have been answered. I wish you every success with this important study. I look forward to the results!

7. PLOS authors have the option to publish the peer review history of their article (what does this mean? ). If published, this will include your full peer review and any attached files.

**Do you want your identity to be public for this peer review?** For information about this choice, including consent withdrawal, please see our Privacy Policy .

Reviewer #1: No

---

## [Editor Report · Acceptance letter]

PONE-D-23-39716R2

PLOS ONE

Dear Dr. Cohen,

I'm pleased to inform you that your manuscript has been deemed suitable for publication in PLOS ONE. Congratulations! Your manuscript is now being handed over to our production team.

Kind regards,

on behalf of

Dr. Nishant Premnath Jaiswal

Academic Editor

PLOS ONE